



# Mean age of stratospheric air derived from AirCore observations

Andreas Engel[1], Harald Bönisch[1,2], Markus Ullrich[1], Robert Sitals[1], Olivier Membrive[3], Francois Danis[3] and Cyril Crevoisier[3]

[1] Institute for Atmospheric and Environmental Science, Goethe University Frankfurt, Frankfurt, Germany

[2] now at Karlsruhe Institute of Technology, KIT, Karlsruhe, Germany.

[3] Laboratoire de Météorologie Dynamique (LMD/IPSL), CNRS, Ecole polytechnique, Université Paris-Saclay, Palaiseau, France

*Correspondence to*: A.Engel, an.engel@iau.uni-frankfurt.de

**Abstract.**

Mean age of stratospheric air can be derived from observations of sufficiently long lived trace gases with approximately linear trends in the troposphere. Mean age can serve as a tracer to investigate stratospheric transport and long term changes in the strength of the overturning Brewer-Dobson circulation of the stratosphere. For this purpose, a low-cost method is required in order to allow for regular observations up to altitudes of about 30 km. Despite the desired low costs, high precision and accuracy are required in order to allow determination of mean age. We present balloon borne AirCore observations from two mid latitude sites: Timmins in Ontario/Canada and Lindenberg in Germany. During the Timmins campaign five AirCores sampled air in parallel from a large stratospheric balloon and were analysed for $CO_2$, $CH_4$ and partly CO. We show that there is good agreement between the different AirCores (better than 0.1%) especially when vertical gradients are small. The measurements from Lindenberg were performed using small low-cost balloons and yielded very comparable results. We have used the observations to extend our long term data set of mean age observations at Northern Hemisphere mid latitudes. The time series now covers more than 40 years and shows a small, statistically not significant positive trend of 0.15±0.18 years/decade. This trend is slightly smaller than the previous estimate of 0.24±0.22 years/decade which was based on observations up to the year 2006. These observations are still in contrast to strong negative trends of mean age as derived from some model calculations.



# 1  Introduction

Mean age of stratospheric air is the average time it takes for the atmosphere to transport air from the tropospheric source region to a given place in the stratosphere. The concept of mean age was first developed by Kida (1983) and has since been refined and discussed in several high quality reviews (Hall and Plumb, 1994;Waugh and Hall, 2002). In brief, the concept divides an air parcels into irreducible fluid elements which are irreversibly mixed during transport in the stratosphere. Each such fluid element has a separate transport time and transport path associated to it. The distribution of the statistical probability associated with the different transit times is called the age spectrum and represents a probability density function (pdf) for individual transit times to this air parcel. The first moment of the age spectrum is called the mean age of air. While the age spectrum cannot be measured, mean age can be derived from observations of inert trace gases under certain conditions. In case of an inert tracer with a perfectly linear trend in the atmosphere, the time lag between the occurrence of a given mixing ratio of a tracer in the troposphere and the occurrence of the same mixing ratio at some place in the stratosphere would be the mean age of air. The two tracers which have been used most widely for this purpose are $CO_2$ and $SF_6$. Neither of these gases increases completely linearly with time, so the shape of the age spectrum needs to be taken into account in deriving mean age.

Mean age has been identified as a valuable tracer to investigate stratospheric transport time scales, e.g. by comparing model derived mean age with observations. Long term trends in mean age have been used to investigate long term changes in the overall overturning circulation of the stratosphere (Brewer-Dobson circulation, BDC). An increase in the strength of the BDC is expected from model calculation, which should be reflected in overall shorter transit times, thus also lower mean age values.

The experimental data base of mean age observations for the verification of such changes is sparse. It relies mainly on very sporadic balloon borne observations of $CO_2$ and $SF_6$ dating back to 1975 (Engel et al., 2009) and on satellite observations of $SF_6$ (Stiller et al., 2012;Haenel et al., 2015). The balloon borne observations used in Engel et al. (2009) were taken in a region between 24 and 35 km where the vertical gradient in mean age at Northern Hemisphere mid latitudes was found to be very small, leading to little variability in this region. The balloon data was limited to a total of 28 flights and showed a positive trend of 0.24 years per decade for this region, which was however estimated to be non-significant. Satellite observations of $SF_6$ used in Stiller et al. (2012) and Haenel et al. (2015) were limited to the lifetime of the Envisat satellite





of about 10 years. They show an uneven distribution of trends with positive trends in the middle
stratosphere of the northern Hemisphere but negative trends in the Southern Hemisphere. Mod-
elling work by Garny et al. (2014) showed that mixing has a strong influence on mean age and
that enhanced mixing leads to higher mean ages in large parts of the stratosphere ("aging by
mixing"). Ploeger et al. (2015) then showed that trends in mean age are to a large degree also
influenced by trends in mixing and not only in residual transport. Overall, it has become clear
that the interpretation of changes in mean age as changes in residual circulation is inadequate,
but rather that it represents a combination of changes in mixing and in residual transport.
The experimental investigation of changes in mean age of stratospheric air is to a large degree
restricted by the availability of observations. The balloon borne data set presented in Engel et
al. (2009) relies to a large part on samples collected in the stratosphere using large and heavy
cryogenic whole air samplers. These instruments require large and expensive balloons to carry
them to altitudes above 25 km. The use of these large balloons involves a large operational team
and is very expensive. The uncontrolled parachute descent of such large payloads after the
flights further presents a large operational constraint due to safety regulations. These safety
regulations make it virtually impossible to fly such large payloads in densely populated areas
as Central Europe. Due to these operational constraints and in order to create a larger and more
representative data base, an easy to launch and cheap technique to allow for the measurement
of age tracers would be required. AirCore, a new technique to sample air which has been sug-
gested by Karion et al. (2010) may provide such an opportunity. In brief, this technique relies
on collecting air in a previously evacuated, long stainless steel tube. When deploying AirCore
on a balloon, the tube, which is open on one side and closed on the other side, is filled with a
fill gas (FG) which has different chemical characteristics from the ambient air to be sampled.
The tube is emptied during ascent of the balloon due to the decreasing pressure with altitude.
Upon descent of the balloon, ambient air is pushed into the AirCore. Due to the length of the
tube and the laminar flow during the collection, the air is only partially mixed and the infor-
mation on the vertical distribution is retained for a while before eventually being mixed due to
molecular diffusion. After collection, the sampled air can then be analysed by pushing it out of
the tube with a push gas (PG), which must again be well distinguishable from ambient air.
A very light weight AirCore developed at University Frankfurt for deployment on small, cheap
and easy to launch balloons as used for launching of e.g. ozone sondes is presented in section
2 together with the analytical set-up for measurements of the AirCore and the data retrieval. In
section 3 we present observations from two mid latitude campaigns, the first one in Timmins,





Ontario in 2015 and the second one in Lindenberg, Germany in 2016. Results from a first test
campaign in Timmins, Ontario in 2014 have been published in Membrive et al. (2016). Due to
technical problems the results from the campaign in 2014 cannot be used to derive mean age.
The mean ages calculated from the observations in 2015 and 2016 are presented in section 4
together with and an updated long term evolution of mean age. Summary and conclusions are
given in section 5.

## 7  2   University Frankfurt AirCore

The AirCore used by University of Frankfurt was developed under two main aspects. The first
aspect is that the instrument should be sufficiently light to allow for flights under simple bal-
loons at mid latitudes in Europe. The second aspect is that the AirCore should be optimised to
allow measurements at high altitudes with an optimal resolution. The AirCore is currently used
for measurements of $CO_2$, $CH_4$ and CO.

### 13  2.1   Overall concept

As explained above, the University Frankfurt AirCore is designed to provide optimum resolu-
tion in the stratosphere while keeping the weight sufficiently low for use under a small balloon.
The vertical resolution, which can be achieved by AirCore measurements, will generally de-
pend on the geometry of the AirCore itself, on the effective volume of the analyser deployed
and on the storage time between collection of the sample and the analysis. The target of our
AirCore is to derive mean age from $CO_2$. As the loss of $CH_4$ in the stratosphere results in the
production of $CO_2$, $CH_4$ needs to be measured simultaneously. We therefore decided to use a
Picarro G2401 analyser for this, which is able to measure $CO_2$, $CH_4$, CO and $H_2O$ with a tem-
poral resolution of about 2-3 seconds and very high precision, which is better than 0.01% for
$CO_2$ and 0.05% for $CH_4$ over a 5 sec period under typical ambient conditions.
Molecular Diffusion is described by Fick's first law of diffusion, which states that the diffusive
flux $J$ is proportional to the concentration gradient $\frac{\partial c}{\partial x}$ and the Diffusion-constant $D$.

$$J = -D \times \frac{\partial c}{\partial x}$$

The same amount of air stored in a short tube with large inner diameter will be stretched out
over a much shorter distance than in a narrow and longer tube. The diffusive flux is thus lower





when using a longer and thinner tube. As the amount of sample collected by AirCore is propor-
tional to the ambient pressure, very little air is collected at high altitudes. In order to minimize
loss of vertical resolution with altitude, it is thus desirable to have thin and long tubes for the
storage of stratospheric air. On the other hand, the absolute amount of air collected by the Air-
Core is limited by the total volume of the tube, which is low for tubing with small inner diam-
eter. We have therefore decided to construct an AirCore from different diameter tubes in such
a way that the high altitude air is stored in the thin diameter tubing part of the AirCore while
the overall volume is provided by wider diameter tubing in which the lower altitude air will
eventually be stored. The AirCore operated at University Frankfurt is thus composed of 20 m
of 8mm O.D. tubing, and 40 m each of 4 and 2 mm outer diameter tubing. The thinnest walled
tubing we could identify were 0.2 mm wall thickness for 8 mm and 4 mm outer diameter tube.
A 2 mm outer diameter tube with an 0.12 mm wall thickness is available. The volume per
weight is highest for the large outer diameter tubing. All tubings were custom produced for the
production of our AirCores. The tubes are joined by light weight adaptors allowing to solder
the tubes together. As suggested in Karion et al. (2010), all tubes were silanized prior to solder-
ing them together. The AirCore is closed during flight on the 2 mm side, while the 8 mm O.D.
tube is the open ended side of AirCore. The calculated weight of the AirCore based on the
specifications of the tubes is 1.4 kg for the 100 m long tube. The final weight of the tube was,
however, slightly higher due to the wall thickness being on the high side of the specified toler-
ance. An automated closure valve (Chen et al., 2017) is added on the closed side and a sample
dryer is mounted on the open end. The drier is based on $Mg(ClO_4)_2$ filled in a ½" O.D. tube of
50 mm length containing a total of 4cm$^3$ of $Mg(ClO_4)_2$. The AirCore is mounted in a Styrofoam
box for thermal insulation and mechanical protection. As the temperature of the AirCore during
the collection determines the amount of air, which can be sampled, we monitor this temperature
with a minimum of three temperature sensors. The automatic closure valve (Chen et al., 2017)
closes the AirCore after landing. This valve is controlled by a light weight electronics package
which also includes the data logger for the temperature sensors and was developed at University
Groningen (Chen et al., 2017). The overall weight of the AirCore in flight mode and using the
University Groningen electronics package and closure valve is about 2.5 kg including a protec-
tive housing.
## 2.2   Estimated vertical resolution
The vertical resolution of the derived mixing ratio profiles is influenced by sampling, storing
and measurements procedures. The principal procedure to estimate vertical resolution has been



outlined by Karion et al. (2010). Resolution is lost due to molecular diffusion during the storage
of the sampled air in the AirCore and due to mixing during the sampling and analysis process.
Molecular diffusion can be calculated using Fick's law. The mixing process is essentially in-
fluenced by two parameters: (i) Taylor dispersion during the collection and analysis of the sam-
ples and (ii) the effective cell volume of the analyser, which has to be flushed. Mixing processes
are species independent, while the first effect (molecular diffusion during storage) is species
dependent, each species having a different molecular diffusion coefficient. The cell volume of
the analyser is not an intrinsic limitation of the resolution of the AirCore itself, but will be
included in the derived resolution based on the Picarro G2401 analyser used for our analysis.
As molecular diffusion is a function of time and molecule, the resolution of our AirCore is also
a function of time and will deteriorate with time and differ for each molecule. As explained
above, molecular diffusion will lead to more loss of resolution in a wider tube. Therefore, our
AirCore loses resolution much stronger at lower altitudes during storage, where the sampled air
is the wider tube. We have applied the same parameters as described in Karion et al. (2010) to
derive the vertical resolution for our AirCore. Figure 1 compares the vertical resolution of our
AirCore to other AirCore systems (Karion et al., 2010;Membrive et al., 2016). The calculation
is based on the assumption of sampling air down to 1000 hPa. At the upper altitudes, the reso-
lution is dominated in these calculation by the effective volume of the analyser cell, while mo-
lecular diffusion is the dominant term at low altitudes. The overall vertical resolution of the
measurements is better than 1 km below 24 km altitude and increases to about 2.5 km at 30 km
altitude (Fig. 1). For comparison, the HR-AirCore described by Membrive et al. (2016) achieves
a vertical resolution which is better than 300 m below 15 km and better than 500 m below 22
km, however using much longer tubing resulting in a higher weight.
**2.3   Operation and analytical setup**
Before the flight the AirCore is checked for leak tightness and cleanliness. As a first test, a gas
of known concentration is measured either directly or by passing it through the AirCore and the
measurements are compared. As a further test, the AirCore is filled with a gas of known con-
centration and analysed again after a storage time of 24 hours. Only if the $CO_2$ and $CH_4$ readings
from both values agree within the uncertainties, the AirCore is considered as clean and leak
tight. It is then filled with a fill gas (FG) of known $CO_2$, $CH_4$ and CO concentration no longer
than 24 hours before flight. Before flight the automatic valve mounted on the inlet side of the
AirCore is opened.



The AirCore should be analysed as quickly after the flight as possible as molecular diffusion
decreases the achievable vertical resolution. During the flight in Timmins, Ontario, this oc-
curred about 4 hours after the ~300 km flight, while the analysis started within an hour after
landing during the flights launched from Lindenberg. In order to achieve this fast analysis, the
analytical setup consisting of a Picarro analyser and a gas control system must be deployed in
the field. For this purpose, we mounted the analytical system in a car when operating from
Timmins and inside our laboratory bus during flights from Lindenberg. The setup also included
a battery-operated inverter allowing to run the Picarro for up to 6 hours, thus also allowing to
keep the instrument heated and under constant flow while driving to the predicted landing area.
We use the same gas as Fill Gas (FG) and Push Gas (PG). This gas-mixture contains typical
atmospheric $CO_2$ values, typical $CH_4$ values expected around 30 km altitude and significantly
higher CO values than observed either in the troposphere or in the stratosphere. Based on the
CO values, it is thus possible to distinguish between the sampled atmospheric air, the PG and
the FG, which is left in the tube.
The gas flow system used for the analysis of AirCore is shown in Figure 2. This system allows
to flush the two lines which are needed to connect the AirCore for the analysis with a standard
gas (Cal Gas) or the push gas used for the analysis (connection and flushing, upper panel in
Figure 2). During the connection all dead volumes of the connectors can be flushed, minimizing
the contamination from ambient air. During the connection the Picarro is flushed with push gas.
The pressure of the PG is regulated to slightly above ambient pressure (typically 1030 hPa) and
the flow through the Picarro is regulated to 40 ml/min. Once all lines are flushed and connected
and the Picarro gives a stable reading of the expected value for the PG, the two-position position
valve can be switched. PG regulated at 1030 hPa is then flushed through the AirCore with a
flow of 40 ml/min. The stratospheric (upper altitude) air is flushed out first. Before stratospheric
air arrives at the Picarro analyser, the standard gas that was used for flushing the connection
lines will arrive first, followed by the remaining FG from the AirCore. The amount of FG left
will depend on the lowest pressure reached during flight and on the temperature of the AirCore
during that phase (see section data retrieval). A typical example of the raw analytical results
from the measurements of two AirCores embarked simultaneously on board the flight from
Timmins in 2015 (see section 3.1) is shown in Figure 3. Figure 4 shows a zoom on the CO
measurements shortly after switching the rotary valve. The PG with its high CO values close to
1.4 ppm is measured. Then the CO values drop due to the lower values of the calibration gas
with which the connection line was flushed. This is then followed by the FG remaining in the
tube. Note that during this flight the amount of FG left was not sufficient for the Picarro to



arrive at its expected value of close to 1.4 ppm. After the passing of the peak with high CO
from the FG, the values drop sharply showing much lower CO values. These lower CO values
are expected in the middle stratosphere (Toon et al., 1999;Engel et al., 2006b) due to the pho-
tochemical balance between production of CO from oxidation of $CH_4$ and the breakdown of
CO due to oxidation with the OH radical. The transition from high CO (remaining FG) to low
CO thus marks the beginning of the sampled air at the upper part of the profile. In a similar
way, the transition from rather low tropospheric CO to high CO marks the time when all the
sampled air has been pushed out of the AirCore and the PG used to push the air out of the
AirCore is seen by the analyser.
## 2.4   Data retrieval
The Picarro analyser will deliver a time series of mixing ratios as a function of measurement
time. The absolute values of the Picarro analyser are transferred to the WMO scale (X2007
scale for $CO_2$ and X 2004a scale for $CH_4$) based on a calibration function derived from absolute
values of 4 gas bottles with a range of $CO_2$ values between 390 and 416 ppm of $CO_2$ and 1.07
to 1.91 ppm of $CH_4$ (only 3 gas bottles). The mixing ratios determined by the Picarro analyser
have to be matched to the altitude at which the air was sampled by the AirCore during the flight.
The basis of this altitude attribution is the ideal gas law and the molar amount sampled at each
altitude during the flight. This matching is achieved in a 4 stage process. First (i) the amount of
remaining Fill Gas is determined, then (ii) the sampling of air based on the ideal gas law is
calculated. In the third step (iii) the start and end times of AirCore in the analyser time series
are determined and finally (iv) the sampling and the analysis can be matched based on the molar
amount.
### 2.4.1   Determination of remaining Fill Gas
In the case of slow vertical displacement of the balloon, pressure equilibrium between the Air-
Core and the surrounding air can be assumed. Under this assumption of an instantaneous pres-
sure equilibrium, the molar amount $n$ of an ideal gas stored in volume $V$ at pressure $p$ and
temperature $T$ is according to the ideal gas law:

$$n = \frac{p \cdot V}{R \cdot T}$$





where R is the general gas constant. The temperature in this aspect is not the ambient tempera-
ture but the temperature of the coil, as we assume an instantaneous equilibrium between tem-
perature of the air inside the AirCore with the coil temperature. In the (i) first step the amount
of FG remaining in the AirCore at the top of the profile is then calculated by searching for the
minimum in $\frac{p}{T}$.
As noted above, the assumptions about pressure equilibrium between air inside the AirCore and
outside air needs to be made in this calculation. While this is certainly a valid assumptions for
a slow descent of the balloon, it will be less valid the faster the descent of the balloon. In the
case of a rubber balloon which will burst while still ascending and then immediately start to
descent the situation is even more difficult, as the pressure inside the AirCore will actually be
higher than the outside pressure during the beginning of the descent, because a non-equilibrium
will exist both for the emptying of the tube during ascent and the re-filling of the tube during
the descent. The size of this non-equilibrium effect will depend on the geometry of the AirCore
but also on the filling of the sample dryer. In particular, this latter may provide a significant
flow restriction if the $Mg(ClO_4)_2$ is packed very densely. The amount of FG left in the AirCore
is thus expected to differ significantly from the equilibrium amount calculated before based on
the minimum in $\frac{p}{T}$. We have taken particular care to have a short and loosely packed dryer
providing minimal flow restriction. As the FG used in our case differs significantly in CO val-
ues from ambient air and from the calibration gas used, it is possible to determine the amount
of remaining FG by integrating the CO peak observed during the measurement of the AirCore
(Chen et al., 2017) as illustrated in Figure 4. When switching from the bypass to the measure-
ments mode, the gas inside the Picarro measurement cell is first replaced by the calibration gas,
which was used to flush the transfer line to the AirCore during the connection. The calibration
gas (Cal Gas) is then replaced by the FG which remained in the AirCore and then by ambient
stratospheric air. All of these gases are partially mixed and all of them contain some CO. In
order to separate the amount of CO due to PG and Cal Gas from the signal due to remaining
FG, we performed a measurement in a similar set-up but with our AirCore filled with pure
nitrogen, which contained no detectable amounts of CO. The CO in this set-up is thus not due
to remaining FG and can be used to correct the offset due to PG and Cal Gas when integrating
the CO peak from the remaining FG. Using the known mixing ratio of CO in the FG, the molar
amount of remaining FG can be determined. In the case of the two fast descent profiles from
Lindenberg (see section 3), the pressure of remaining FG was determined to be 17.2 hPa and
7.3 hPa, respectively, while the corresponding pressures derived from the minimum in $\frac{p}{T}$ was





slightly lower at 15.2 hPa and 7 hPa, respectively. The differences are rather small, correspond-
ing to altitude differences of a km or less, as the flow restriction of our AirCore is low due to
the large inner diameter of the 8 mm tube (which carries the largest part of the volume) and the
carefully packed sample dryer. In the following we have therefore only corrected this effect by
adopting the upper sampling pressure during the Lindenberg flights to the value calculated from
the integration of the CO peak of the remaining FG.

### 7 *2.4.2 Sampling of ambient air with AirCore*

In the second step, we determine the amount of moles sampled during every time step of the
balloon trajectory. Again in case of slow descent the assumption of pressure equilibrium be-
tween the tube and the sampled ambient air is justified. Starting at the molar amount determined
during step 1 and adding up over the pressure and coil temperature measured during the descent
of the balloon results in a matrix linking ambient pressure and altitude to sampled molar
amount. In the case of a faster descent, the assumption of pressure equilibrium is not completely
valid, but as shown in section 2.4.1. the effect is small. To a first order this is compensated by
starting the summing at the molar amount determined from integration of the CO peak as de-
scribed in section 2.4.1.
The approach of summing up the amount of moles in the AirCore during the flight will take
also into account that air from the AirCore can be lost again, if the pressure of ambient air is
below that of the AirCore, e.g. in case the balloon ascends (which can occur for large strato-
spheric balloons) or if the AirCore heats up after landing without being closed. The procedure
is thus integrated in time until the moment that the AirCore is closed either manually or via an
automatic closure valve.

### 23 *2.4.3 Matching AirCore and Picarro data*

In the third step we determine the starting and ending point in the measurements of the air mass
sampled and stored in the AirCore with the Picarro (see Figure 2). This is achieved by fitting a
Gaussian curve to the CO peak from the remaining FG (see Figure 4). In case that there is so
much FG left that the CO peak reaches a plateau the left and right side of the remaining CO
from FG are fitted separately using only one side of the Gaussian for fitting. The time when the
peak reaches half its height is chosen as the start time. The time derived for the rising CO peak
is then taken as the start time of the FG, the time derived for the descending part of the peak is
taken as the start of the AirCore, i.e. ambient stratospheric air. We chose to use this second





point as starting point and associate it with the amount of remaining FG determined in section
2.4.1. The determination of starting and ending point of the AirCore analysis in the Picarro time
series is critical in the correct assignment of the measured mixing ratio to the sampling location
and altitude. Especially the starting point of the AirCore is a critical parameter as an offset of 1
hPa will result in a significant shift in altitude in the stratosphere. The fourth step is then to
calculate the molar amount of air passing through the Picarro and linking this to the molar
amount sampled with the AirCore. The link between the molar amount and the time of meas-
urement is straight forward, as the flow through the Picarro is regulated to be constant (in our
case 40 ml/min) and also temperature and pressure of the measurement cell of the Picarro ana-
lyser are controlled

### 2.4.4  Correction of mixing ratios for mixing between AirCore and fill-gas

In order to keep the effect of mixing between FG and ambient stratospheric air small, we have
made the FG in order to have values close to those expected in the middle stratosphere. The
FG, which is also used as PG has a mixing ratio of about 407.75 ppm of $CO_2$ and 1228.6 ppb
of $CH_4$. On the other hand, the PG has much higher CO (about 1400 ppb) in order to allow a
clear distinction from both tropospheric and stratospheric air. Especially stratospheric air has
much lower CO mixing ratios, which are on the order of 20 ppb (Engel et al., 2006b;Toon et
al., 1999). Our measurements showed a gradual decrease of CO values from the high FG values
to the significantly lower stratospheric mixing ratios. Values as low as 20 ppb where only ob-
served sporadically, with values around 100 hPa pressure altitude typically being in the 30 ppb
range. This enhancement could either be caused by CO production from the reaction of ozone
with the tubing or it could be a measurement artefact as the Picarro is not well suited for such
low CO mixing ratios. This gradual decrease from the FG values to stratospheric values is due
to a combination of mixing and diffusion. While mixing is similar for all species, molecular
diffusion depends on the diffusion coefficient and is different for each gas. The upper part of
the profile is stored in the 2 mm O.D. tube. In this tube molecular diffusion only leads to a very
gradual mixing of the two gases (FG and ambient air). Most of the gradient is due to mixing
during the analysis (the limiting part is the volume of the analyser cell). Therefore, this gradient
can be treated as a gradient caused by mixing and not diffusion, and the mixing should be sim-
ilar for all species. We will thus use the large difference in CO to characterise the fraction of
FG in the analysis and correct the observed mixing ratios of $CH_4$ and $CO_2$ for the remaining
impact of FG. In order to determine the fraction of remaining FG an assumption on the expected



stratospheric mixing ratio of CO must be made. As we expect that the correction should ap-
proach zero once the cell has been flushed a few times, we have chosen to use the average CO
value observed between 80 and 100 hPa as the expected value. Using this target value the frac-
tion of FG is calculated from the difference between the measured and the expected CO mixing
ratios and the observed mixing ratio from the Picarro measurements is corrected accordingly.
## 3   Atmospheric observations
The AirCore developed at University Frankfurt is sufficiently light to be flown with a small
balloon. However, we performed our first test flights using large stratospheric balloons
launched by CNES from Timmins in Ontario. Two such test flights were performed in order to
compare our results with those of other groups. The first test flights were performed in 2014.
The results from the first flight in 2014 is reported by Membrive et al. (2016). Due to a balloon
trajectory which was not adapted to AirCore measurements (long ceiling and long float of the
balloon around 20 km altitude) the profiles obtained for $CO_2$ from AirCore could not be used
to derive mean age, as the AirCore showed unrealistically low values of $CO_2$ around the float
altitude, possibly due to an interference with the sample dryer (Membrive et al., 2016). These
data are therefore not discussed in this paper. During a second flight of the same payload as
reported in Membrive et al. (2016), the vertical velocities were much better adapted and we
could derive profiles of $CO_2$, CO and $CH_4$. These are presented in section 3.1. In section 3.2.
we present the first results from our AirCore measurements at mid latitudes using a small and
easy to launch rubber balloon of similar type as used for ozone soundings.
### 3.1   Timmins 2015
The payload launched from Timmins in the year 2015 was very similar to the one described in
Membrive et al. (2016). It consisted of a combination of two AirCores by University Frankfurt,
one high-resolution AirCore (Membrive et al., 2016) and two light weight AirCores by the
Laboratoire de Meteorologie Dynamique (LMD).  In contrast to the Picarro G2401 used by
University Frankfurt, the LMD team used a G2301 analyser, which lacks the capacity to meas-
ure CO. The payload also included two pico-SDLA spectrometers (Ghysels et al., 2014;Durry
and Hauchecorne, 2005) for measurements of $CO_2$ and $CH_4$, which are based on in-situ infrared
absorption measurements. Results from these latter measurements were perturbed due to ther-
mal drifts in laser emission wavelength and are not available at the time of writing. The balloon
was launched from Timmins, Ontario, on August 22, 2015 and reached a minimum pressure of





about 11 hPa. In order to reach a zone where a safe landing was possible, the balloon was left
to drift westwards and a slow descent of the balloon was started in the early morning of August
23 (8:30 U.T.). The payload was separated from the balloon just below 100 hPa pressure and
the payload landed at about 10:40 U.T. The recovery team was able to recover the payload such
that the analysis could be started about 4 hours after landing.
Figure 5-7 show the vertical profiles of CO, $CH_4$ and $CO_2$ as measured with the Picarro analyser
for both AirCores by GUF and in comparison to the LMD AirCores (only for $CH_4$ and $CO_2$).
The altitude attribution is based on the CO peak as described in section 2.4. One of the AirCores
(AC-2) was operated with the automatic closure valve and thus did not lose any air while warm-
ing up on the ground after landing. The profile from this AirCore extends to the ground, while
other profile (AC-3) ends higher up due to the loss of air. First of all, two peaks in CO are
observed in the troposphere, which are found at the same altitude for both AirCores. These
could have been caused by biomass burning from wildfires occurring over western Canada dur-
ing the period of observations. Lowest values of CO in the stratosphere are on the order of 10-
20 ppb, in agreement with expected steady state values (Toon et al., 1999;Engel et al., 2006b).
AC-3, which was measured after AC- 2 shows an increase in CO mixing ratios above 20 km,
which is most probably due to the longer storage time, resulting in more diffusive mixing with
remaining Fill Gas.
Figure 6 shows the vertical profiles of $CH_4$ derived from the 5 independent AirCores all
mounted on the same gondola. A remarkably good agreement is observed, as already shown for
observations in 2014 (Membrive et al., 2016). As also discussed in (Membrive et al., 2016) it
is obvious that the AirCore-HR is able to capture fine scale vertical structures which are not
present in the profiles derived from the light weight AirCore of University Frankfurt nor from
the light weight AirCore of LMD, which have a similar vertical resolution in the troposphere
but less vertical resolution in the stratosphere. The vertical profile of the light weight LMD
AirCore is only derived up to about 23 km altitude. Above 19 km altitude the light weight LMD
AirCore shows some deviations from the High Resolution AirCore and the University Frankfurt
AirCores. The University Frankfurt AirCore on the other hand is capable of capturing some
local structure around 21-22 km altitude, although the two local minima from the High Reso-
lution AirCore are smeared out to one broader minimum. Above 23 km altitude there seems to
be a small altitude mismatch between the University Frankfurt and the High Resolution Air-
Core, which is however less than 1 km. This altitude discrepancy is explained by the uncertainty
in matching the Picarro measurements to the AirCore sampling, which is also treated slightly



differently in the LMD and the University Frankfurt retrieval. In order to compare the values
of the different AirCores, we binned the data into 1 km intervals and then calculated averages
for each AirCore in these bins. In the troposphere (values between 3 and 13 km altitude) the
standard deviation between these 1 km bins is 1.4 ppb, or 0.08%. In the stratosphere the devia-
tions are higher due to the large vertical gradient. Absolute deviations are on average (between
15 and 24 km altitude) 11 ppb or 0.75%. The agreement of the two University Frankfurt Air-
Cores is much better (0.17 ppb or 0.001% in the troposphere and 3.8 ppb or 0.25% in the strat-
osphere), as is that of the two light weight LMD AirCores (0.28 ppb or 0.0015% in the tropo-
sphere and 1.6 ppb or 0.1% in the stratosphere).
The $CO_2$ measurements from the five AirCore are compared in Figure 7. The overall shapes of
the profiles from the different AirCores show good agreement. In particular also rather small
scale phenomena are resolved and observed in all AirCores. For instance the small scale struc-
ture at around 13 km is observed in all AirCores, again showing that the sampling and the
altitude attribution gives consistent results. As already discussed by Membrive et al. (2016),
$CO_2$ measurements seem to show more deviations. However, it should be noticed that the range
shown for $CO_2$ is much smaller than for $CH_4$. In contrast to $CH_4$ the deviations in the tropo-
sphere and the stratosphere are very similar, as the vertical gradient is similar. In absolute terms,
the deviations are typically 0.35 ppm or about 0.09%. This deviation is thus on a very similar
level as observed for $CH_4$ in the troposphere. Overall, this agreement is very good, taking into
account that the different AirCores partly use different data retrieval algorithms and have dif-
ferent geometries and thus also different vertical resolutions. As in the case of $CH_4$, we note
that the agreement between the two University Frankfurt AirCores is much better (0.04 ppm or
0.01% in the troposphere and 0.17 ppm or 0.04% in the stratosphere), as is the agreement be-
tween the two light weight LMD AirCores (0.05 ppm or 0.015% in the troposphere and 0.07
ppm or 0.02% in the stratosphere). This shows that the differences are systematic and must be
related to the geometries of the different AirCores and the related uncertainties in the altitude
attributions.
**3.2   Lindenberg 2016**
A first test campaign to study the use of our AirCore using small balloons has been conducted
from the Lindenberg Meteorological Observatory, Germany. AirCores were launched on May
20, 2016 and May 25, 2016. The balloon used for the first flight was a TA 1500 balloon. A
larger balloon (TA 3000) was used for the flight on May 25, thus allowing to reach a higher





ceiling altitude. Ceiling pressures were 15.2 and 7 hPa, respectively. Large parachutes were
used in order to slow down the descent speed and minimize the effects due to non-equilibrium
of pressure inside of the AirCore and outside pressure. For both measurement flights we were
able to recover the AirCore very fast and start the analysis within an hour after landing. The
retrieval procedure was similar to the one for the flight from Timmins in 2015 with the excep-
tion that we derived the pressure at which sampling began not from the measurements of am-
bient pressure but from the integration of the CO peak as described in section 2.4.1. (diploma
thesis Markus Ullrich, University Frankfurt, Dec. 2016).
Figures 8 to 10 show the vertical profiles of CO, $CH_4$ and $CO_2$ from the two flights conducted
in May 2016. For CO the general agreement between both measurement flights is very good,
even though they are five days apart. CO values are higher than observed in Timmins 2015.
This could either be due to the use of a different Picarro analyser (note that these values are
close to the detection limits of CO) or to enhanced CO in early spring e.g. due to descending
mesospheric air during the polar winter. For all species there is a distinct change at the tropo-
pause around 11 km altitude with a sharp drop in mixing ratios. The decrease in tracer mixing
ratios is observed at the same altitude as the thermal tropopause, i.e. at 11 km altitude, showing
that the altitude attribution as explained in section 2 yields realistic results.
For $CH_4$ and CO this is due to the chemical loss in the stratosphere, whereas $CO_2$ is very long
lived in the stratosphere. The decrease in $CO_2$ values above the tropopause is mainly caused by
the high values of $CO_2$ in the northern hemisphere troposphere during spring, while the air
above the tropopause partly entered through the tropical tropopause and partly during late sum-
mer of the preceding year when tropospheric $CO_2$ values were lower due to the seasonal cycle
(Boenisch et al., 2009). Both $CH_4$ and $CO_2$ show some fine structures in the stratosphere during
both flights. There is a local maximum in $CO_2$ and $CH_4$ at around 21 km altitude on May 20
and a similar local maximum is observed on May 25 at about 20.5 km altitude. The maxima
and minima in $CO_2$ and $CH_4$ are collocated at the same altitude. Therefore, this is clearly a
dynamical feature where $CO_2$ rich (younger) air (see section 4) is advected which at the same
time has higher $CH_4$ mixing ratios. Such air masses would be expected to occur in the tropics
or subtropics. As the dynamical interpretation of the profiles is not the focus of this paper, this
is not investigated further e.g. by using meteorological data.





## 4   Age of air from AirCore

The main aim of our AirCore activities is to determine mean age of air and use this to extend
our long time series of mean age from balloon observations (Engel et al., 2009). The two tracers
most commonly used to derive mean age are $CO_2$ and $SF_6$. As shown in Engel et al. (2009), the
vertical gradient of mean age becomes rather small at pressure altitudes above 30 hPa. The
mean value of mean age above this altitude has been used to investigate long term changes in
mean age and in the stratospheric circulation (Engel et al., 2009). An ideal tracer for the deri-
vations of mean age should have neither sinks nor sources in the middle atmosphere and show
a monotonous, linear trend in the lower atmosphere (Hall and Plumb, 1994;Waugh and Hall,
2002). Neither $CO_2$ nor $SF_6$ completely fulfil these requirements (Engel et al., 2009) leading to
uncertainties in the mean age values derived from observations. In the case of $CO_2$ there are
three specific issues which need to be considered: (i) the source of $CO_2$ in the middle atmos-
phere due to the oxidation of $CH_4$, (ii) the seasonal cycle of $CO_2$ and (iii) the deviation of the
deseasonalized long term trend of $CO_2$ in the troposphere from linearity. The procedure to cal-
culate mean age and how to take these issues into account is the same as in Engel et al. (2009)
and only briefly summarized here. As $CH_4$ is oxidised in the stratosphere and thus provides a
source for $CO_2$ in the stratosphere, the amount of $CO_2$ produced from the oxidation has to be
subtracted from the observed $CO_2$ mixing ratio. The $CO_2$ produced in the stratosphere is derived
from the observed $CH_4$ by taking the difference to the deseasonalized tropospheric $CH_4$ at the
time of measurements. In this procedure the fact that $CH_4$ has a tropospheric trend and takes
some time to propagate to the stratosphere is ignored. The error in mean age due to this simpli-
fication is less than half a month. $CO_2$ has a seasonal cycle in the troposphere which can prop-
agate into the lower stratosphere (Andrews et al., 2001a;Andrews et al., 2001b;Hintsa et al.,
1998;Boenisch et al., 2009;Engel et al., 2006a). Rosenlof et al. (1997) found that the seasonal
cycle in water vapour is observable up to potential temperatures of about 450 K and termed this
region the tropically controlled transition layer. In the stratospheric overworld (above 450 K
potential temperature) short term influences e.g. due to seasonal cycles in the troposphere or
tropopause region are much smaller. $CO_2$ can thus only be used as an age tracer for air at po-
tential temperatures above 450 K where the mixing ratios are not influenced by seasonality in
the troposphere anymore. Our analysis of mean age is thus restricted to potential temperatures
above 450 K. Thirdly, the deseasonalized tropospheric trend of $CO_2$ in the troposphere deviates
from a perfect linear increase. The mean age derived from $CO_2$ observations will thus depend
on the shape of the age spectrum. To compensate for the effects of this deviation on the mean





age values derived, we again followed the same approach as in Engel et al. (2009). We use a
parameterization of the width of the age spectrum Δ as function of mean age Γ as suggested by
Hall and Plumb (1994), i.e. $\frac{\Delta^2}{\Gamma} = 0.7\ years$ with the general shape of the age spectrum being
an inverse Gaussian function. We have further adapted the fitting period for the tropospheric
trend so as to represent 98% of the air input for each individual data point (i.e. shorter time
periods for the fit are applied for younger air), in order to find the best possible description of
the tropospheric input time series.
The influence on all three effects on the mean age values has been included in the error analysis,
again following Engel et al. (2009).

## 4.1   Vertical profile observations

Figure 11 shows the mean age profiles for the two flights from Lindenberg in May 2016 and
the two AirCores flown simultaneously in August 2015 from Timmins. The data have been
filtered to exclude air masses with potential temperature below 450 K where the $CO_2$ seasonal
cycle is still expected to have a significant impact. As for many other profile observations of
mean age (Andrews et al., 2001a;Engel et al., 2009;Schmidt and Khedim, 1991) a decrease of
mean age with altitude is observed up to about 23-24 km altitude, above which the vertical
gradient becomes very small. Mean age values above this layer are on the order of 5 years, in
very good agreement with other long term data sets. The observations from Timmins in August
2015 show slightly higher mean age values than the observations in May 2016 from Lindenberg
in Germany. This could be explained by the seasonal cycle in mean age derived from MIPAS
Envisat observations of $SF_6$, showing youngest air in the northern hemisphere mid latitudes
above 25 km during winter and oldest air during summer (Stiller et al., 2012). The younger
spring measurements from Lindenberg could thus still be influenced from the lower mean age
values during winter, while the older observations from Timmins in August 2015 should be
during the maximum of the seasonal cycle.

## 4.2   Extension of long term time series

As explained above, we want to use AirCore observation to extend our long time series of
stratospheric mean age observations (Engel et al., 2009). The calculation of a temporal trend in
mean age is complicated by the sparsity of the data set in combination with a vertical gradient



in mean age. As the AirCore observations, in agreement with other balloon data, only show a
very small vertical gradient above an altitude of 23-24 km, corresponding to about 30 hPa, we
have adopted the same procedure as used in Engel et al. (2009), i.e. we average all data between
30 hPa and and 5 hPa and calculate an average value of mean age for this region of the strato-
sphere.
The last data point in Engel et al. (2009) was from the year 2005. There is thus a gap of 10 years
between these last measurements and the new AirCore data. Overall the mean values derived
from the AirCore data are in very good agreement with the values published in Engel et al.
(2009). The mean values for mean age above 30 hPa from the Timmins flights are $4.9\pm1$ and
$5.3\pm1$ years, while the Lindenberg flights yield slightly lower mean age values at $4.7\pm1$ and
$4.8\pm1$ years. Note that the uncertainty ranges include uncertainties in the estimated representa-
tiveness, the derivation of mean age and the observations themselves, based on the error assess-
ment given in Engel et al. (2009). Using all the data available, we derive a new estimate of the
long term trend in mean age for the mid latitude stratosphere of the Northern Hemisphere be-
tween 30 and 5 hPa. The time period covered is now more than 40 years (1975-2016), albeit
with a very restricted number of profile observations. The updated trend is now calculated to
be $0.15\pm0.18$ years/decade. This trend is smaller than the previous estimate ($0.24\pm0.22$
years/decade), but agrees within the uncertainty range. The positive trend is not significant
within the one sigma uncertainty range. Due to the reduced uncertainty of the new estimate, the
largest negative trend which would be compatible with our data within the two sigma uncer-
tainty range remains nearly unchanged (-0.21 years/decade instead of -0.2 years /decade).

## 5   Summary and conclusion

Observations of stratospheric trace gases are well suited to investigate chemical and physical
processes in the stratosphere. They are also well suited to investigate long term changes in the
stratosphere. In particular, for the investigation of long term changes, high precision and accu-
racy are needed in combination with rather low costs (Müller et al., 2016;Moore et al., 2014).
Most in-situ measurements in the stratosphere require the use of large and expensive balloons
to carry instruments above altitudes of 20 km. The new technique of AirCore (Karion et al.,
2010;Membrive et al., 2016) is ideally suited to provide such low cost long term observations.
We have thus investigated the usefulness of AirCore for investigations of long term evolution
of mean age in order to extend the existing balloon data set used in Engel et al. (2009) and Ray
et al. (2010). University Frankfurt has developed an AirCore system from three different inner



diameter tubes, targeted at an optimal vertical resolution in the stratosphere, while still being
sufficiently light weight to be deployed on a small balloon. During an intercomparison cam-
paign in Timmins, Ontario in August 2015 we compared 5 different AirCores. We have shown
that the AirCore technique can be used to derive high precision vertical profiles of $CO_2$ and
$CH_4$. Both LMD and University Frankfurt measurements are referenced to the same scales, but
using independent calibrations. This shows that the results are also of high accuracy. For both
trace gases the comparison between the different independent AirCores was better than 0.1%
when vertical gradients are small, as is the case for $CO_2$ in the mid latitude stratosphere above
a pressure altitude of 30 hPa (about 24 km altitude) and for tropospheric $CH_4$. This indicates
that the For $CH_4$ in the stratosphere, where there is a large vertical gradient the typical agree-
ment was still better than 1%. The agreement between two similar AirCores, which sampled in
parallel was always better than the agreement between different AirCores when sampling in
parallel. This shows that the geometry, the analysis system and the data retrieval of the AirCore
has a significant impact. We have further performed first observation from the mid latitude site
of Lindenberg in Germany using small rubber balloons. Due to careful planning, it was possible
to analyse the AirCores within one hour after landing. We showed that the CO peak from the
remaining Fill Gas in the AirCore can be used to derive the maximum sampling altitude. In the
case of the observations from Lindenberg the maximum sampling altitudes were only slightly
lower than the maximum pressure altitude of the balloon. This is a good indication that the flow
restriction was rather small and that the pressure equilibrium between the tube and outside air
is rather fast. This was achieved in particular due to the use of a large inner diameter tube for
the main volume of the AirCore and by using a sample dryer which was optimised for minimum
flow restriction. Nevertheless, the altitude attribution of the sampled air remains a difficult is-
sue, in particular when descent rates are high. The use of balloon techniques allowing for rather
slow descents should thus be considered when setting up AirCore measurements sites.
We have used the new observations to calculate mean age of stratospheric air. The results from
our AirCore observations are in very good agreement with previous observations using whole
air sampling techniques (Engel et al., 2009) with values ranging from 4.7 to 5.3 years of mean
age above a pressure altitude of 30 hPa. We have used these data to extend our long term time
series of balloon borne mean age observations. This time series now dates from 1975 to 2016,
thus spanning a total of more than 40 years. The long term trend of mean age in the northern
hemisphere mid latitude stratosphere deduced from this data set is 0.15±0.18 years/decade. This
trend is smaller than the previous estimate (0.24±0.22) years/decade but remains well within
the uncertainty limit. Based on this analysis, we thus sustain our result that no significant change



in mean age of air for the mid latitude stratosphere of the Northern Hemisphere can be derived
from our data set. Despite the smaller positive trend derived from this extended data set, large
negative changes in mean age in this region can still be excluded, as the uncertainty on the
derived trend has been reduced. A negative trend in mean age of more than -0.2 years/decade
for the middle stratosphere of the Northern Hemisphere mid latitudes can still be excluded with
95% confidence.
We conclude that we have shown that AirCore measurements even when using small balloons
can be used to derive vertical profiles of $CO_2$ and $CH_4$. These observations are of sufficient
quality to derive mean age of air and use this to extend the currently available data set of strat-
ospheric mean age observations. We suggest that long term observations using AirCore from a
few selected stations covering different latitude bands may provide a useful tool to investigate
long term changes in mean age. An extension of the AirCore technique to other tracers gases
as suggested by (Moore et al., 2014) may provide a valuable addition as these gases can also be
used to study long term changes in the stratosphere. Müller et al. (2016) suggested that a long
term network for water vapour measurements in the stratosphere should be set up to monitor
this radiatively important trace gas. We suggest that such a network could be complemented by
AirCore observations. Such additional AirCore observations would put the observations of
changing water vapour into the general content of a changing stratospheric circulation. In addi-
tion, if both measurements are performed simultaneously, the observations of Methane from
the AirCore instruments and H2O from the water vapour network could be used to derive the
sum of $2*CH_4 + H_2O$, which has been identified to show much less variability in the strato-
sphere than $H_2O$.

## Acknowledgements

The work of University Frankfurt on AirCore has been funded through the ROMIC programme
of the German Ministry of Science and Education (Grant Nr. 01LG1221) and the EU Infra-
structure Project RINGO (Grant agreement. 730944). We would like to thank the French Space
Agency CNES for balloon operations in Timmins and the team of Ruud Dirksen from the Ger-
man Weather Service (DWD) in Lindenberg. The support of the workshops and technicians at
University Frankfurt is gratefully acknowledged. Special Thanks go to Huilin Chen from Uni-
versity of Groningen in the Netherlands for many valuable discussions on AirCore techniques
and an introduction to AirCore measurements during measurements from Sodankylä. Olivier
Membrive was funded by EIT/Climate-KIC, a body of the European Union.





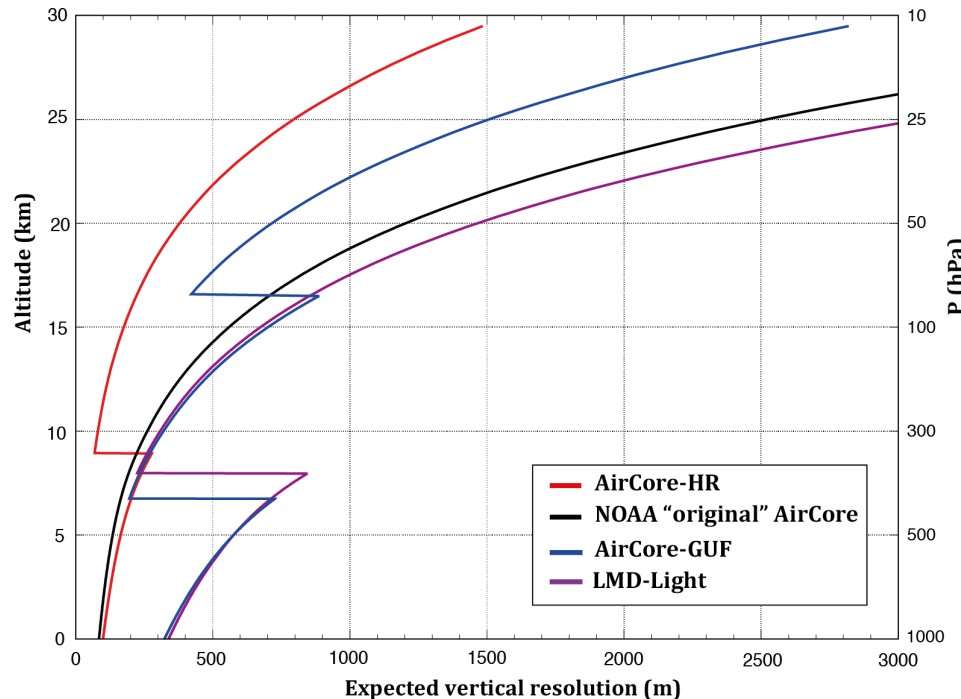

Figure 1: Calculated vertical resolution for $CO_2$ with the Goethe University Frankfurt (GUF)
AirCore (AirCore-GUF) in comparision to other AirCores (see text for description), assuming
a delay of 3 hours between collection and sampling, a measurements flow of 40 ml/min and an
effective cell volume of 6 ml. The AirCore-HR and LMD-Light Aircores are operated by LMD,
while the Noaa "original" AirCore refers to the AirCore described by Karion et al. (2010).



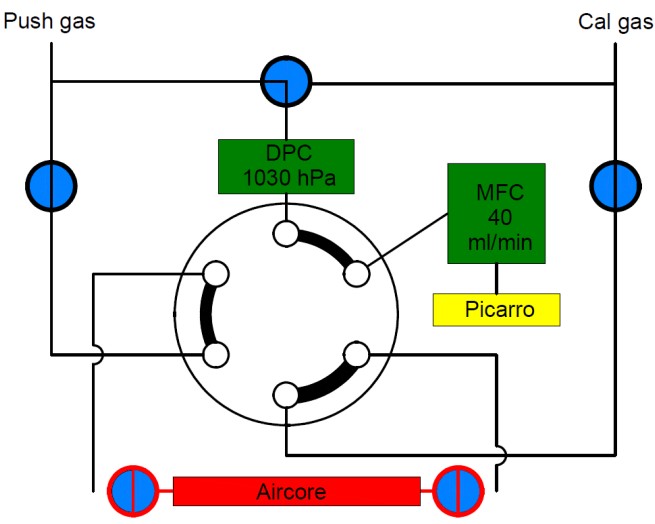

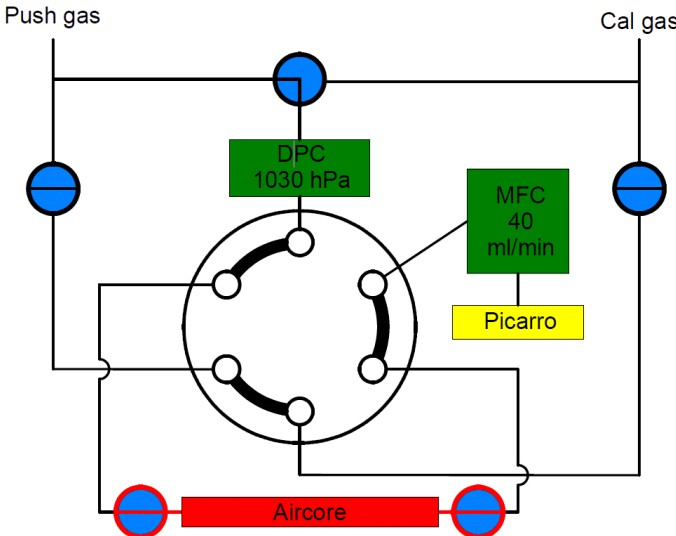

Figure 2: Analytical Setup for the measurement of AirCore. The AirCore including the valves
which are flown is shown in red. In the Bypass/Flushing position (upper panel) the Push Gas
(PG) is measured bypassing the AirCore. The transfer lines to the AirCore can be flushed with
PG or with a calibration standard (Cal Gas), allowing to connect the AirCore to the analytical
system without contamination. For the analysis the transfer lines to the AirCore are closed, the
AirCore valves are opened and the two position valve is switched to the Measurements mode
(lower panel). The PG is passed through the AirCore and pushes the air to the Picarro. Pressure
and flow are controlled allowing for a very constant air flow.





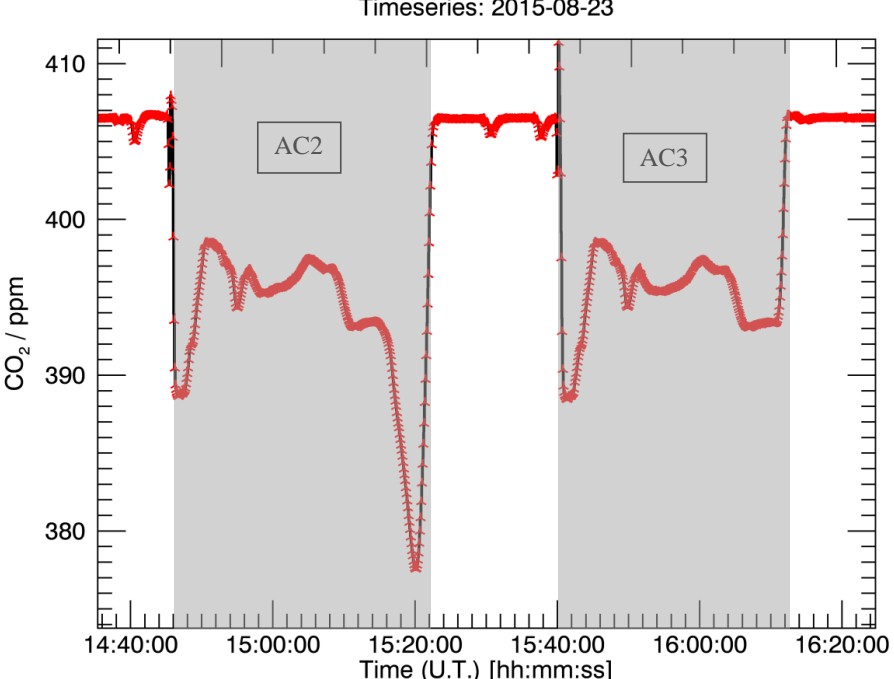

Figure 3: raw analytical results for $CO_2$ from the measurements of the two AirCores flow from
Timmins in 2015. Note that AC 2, which was measured first, contained a closure valve, while
AC 3 was left open until the recovery team was able to reach the instrument. The lowest part
of the AC 3 profile was therefore lost due to the warming of the AirCore on ground.



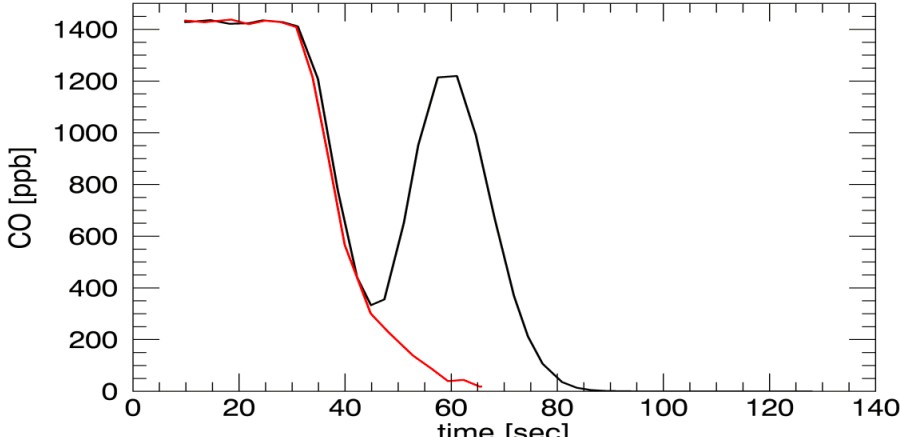

Figure 4: zoom on CO measurements from the flight on May 25, 2016. After switching the two
position valves to measurement mode. Before switching, the analyser measures the high CO
mixing ratios of the PG, then a decrease in CO is observed, representing the low CO values of
the Cal gas used for flushing the transfer lines, and finally the CO peak from the remaining FG
(note that PG and FG are taken from the same gas cylinder and thus have identical mixing
ratios) is measured. After the remaining PG has passed the analyser, the CO values drop to the
expected low stratospheric CO values. The red line shows the CO values measured using the
same setup but analyzing an AirCore filled with CO-free nitrogen. The area between the black
CO peak and the red baseline represents the amount of FG left in the AirCore.



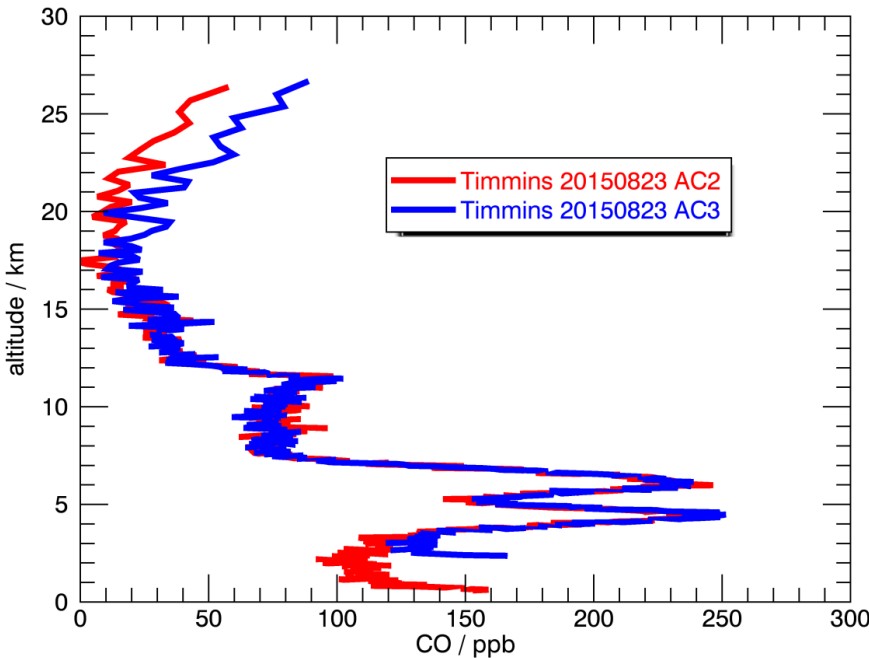

Figure 5: Vertical profiles of CO derived for the flight on August 23 2015 from Timmins, On-
tario, Canada. The two peaks with enhanced CO around 5 km altitude are observed in both
AirCores and are probably caused by wild fires occurring in Canada during August 2015. There
are no CO measurements from the LMD AirCores due to the Picarro analyzer used by LMD.





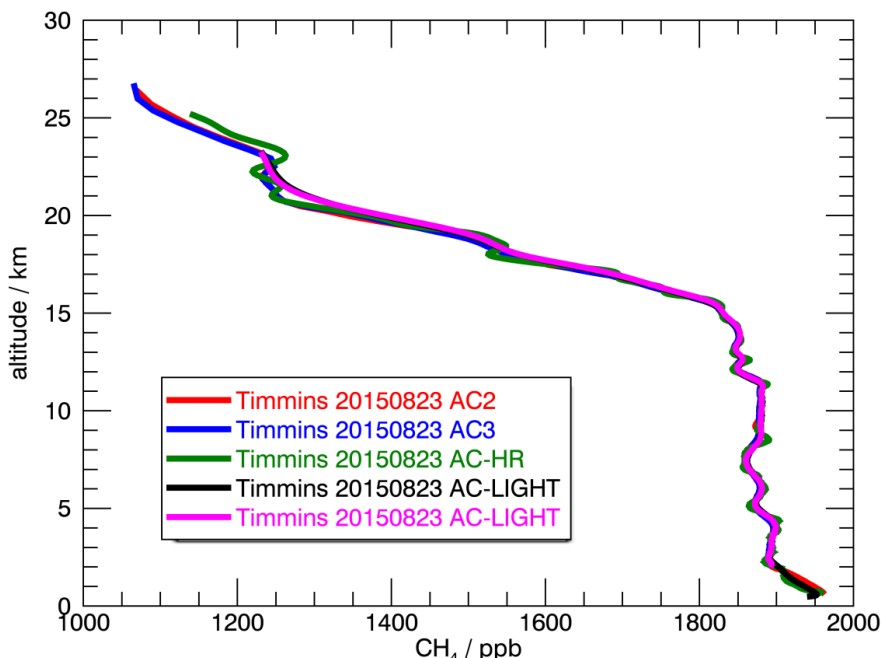

Figure 6: Vertical profiles of $CH_4$ derived for the flight on August 23 2015 from Timmins,
Ontario, Canada. Excellent agreement is observed between all AirCores. There is a slight high
bias of the ligh LMD AirCores above 20 km with respect to both the University Frankfurt Air-
Cores (red and blue trace) and the high resolution (HR) AirCore (green trace). The fine struc-
tures observed by the HR AirCore are smeared out in the light weight AirCores by University
Frankfurt and LMD





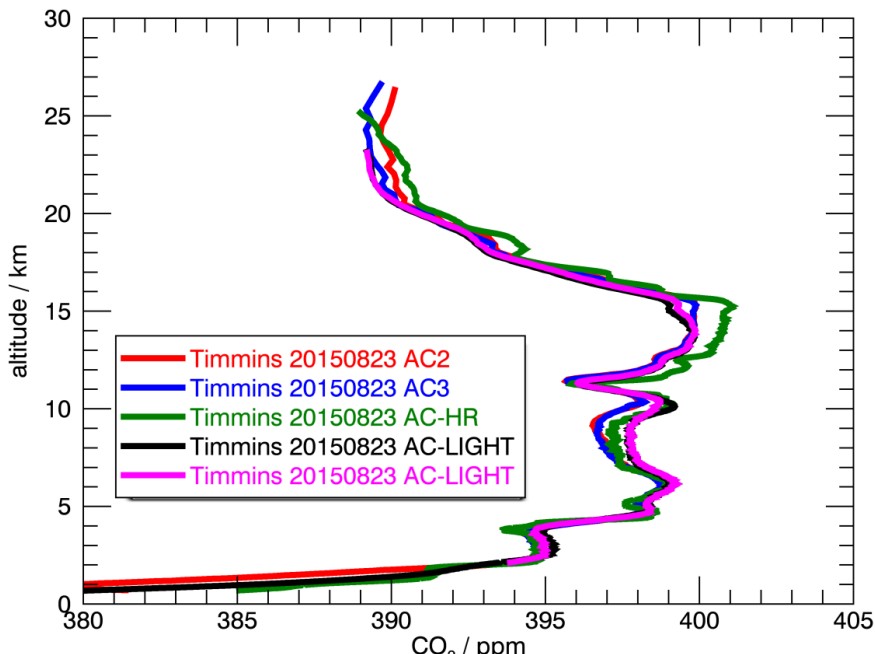

Figure 7: Vertical profiles of $CO_2$ derived for the flight on August 23 2015 from Timmins,
Ontario, Canada. The overall structure are captured very well by all AirCores. Again, more fine
structure is obvious in high resolution AirCore. See text for discussion of the differences be-
tween the different AirCores.





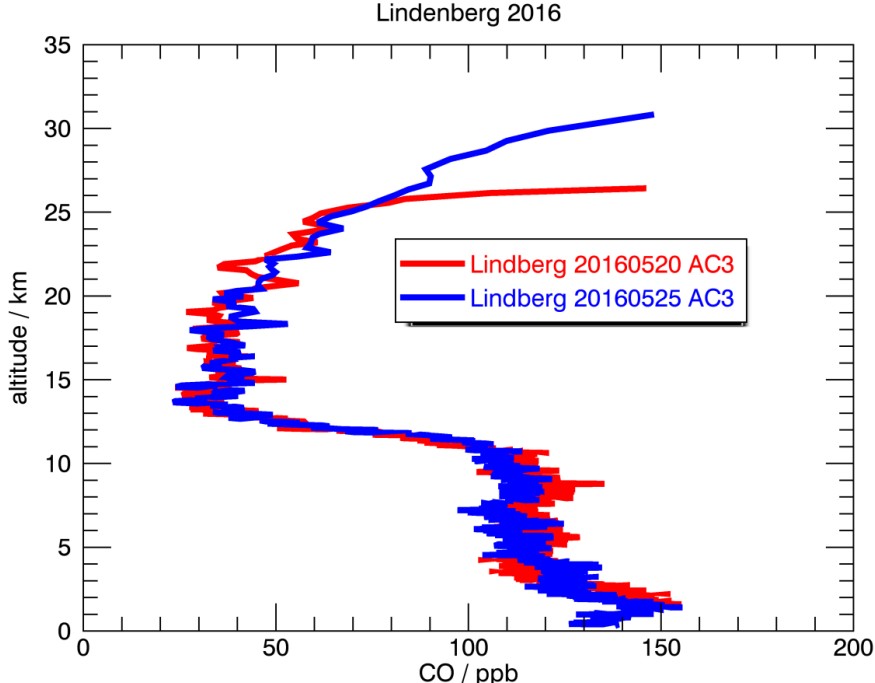

2    Figure 8: Vertical profiles of CO derived for the flights on May 20 and may 25 2016 from

3    Lindenberg in Germany. The flight on May 25 reached higher altitudes due to a larger balloon.





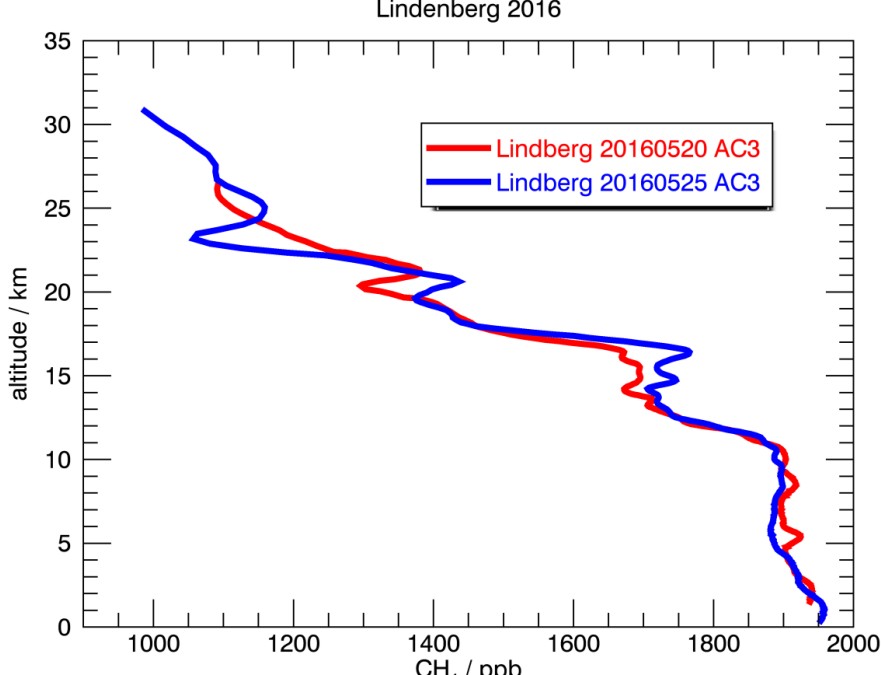

2  Figure 9: Vertical profiles of $CH_4$ derived for the flights on May 20 and May 25 2016 from

3  Lindenberg in Germany. The flight on May 25 reached higher altitudes due to a larger balloon.





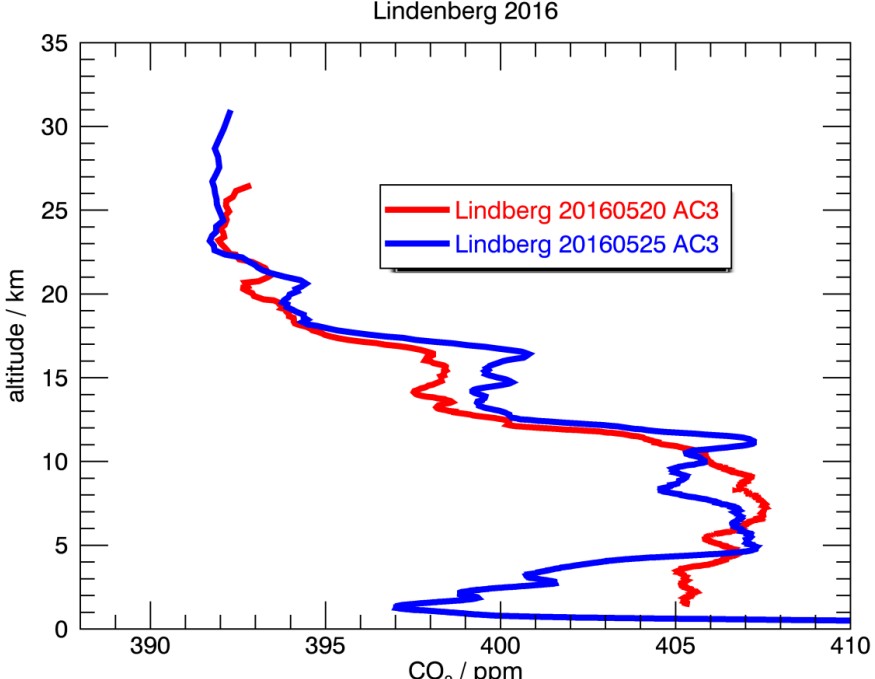

2     Figure 10: Vertical profiles of $CO_2$ derived for the flights on May 20 and may 25 2016 from

3     Lindenberg in Germany. The flight on May 25 reached higher altitudes due to a larger balloon.




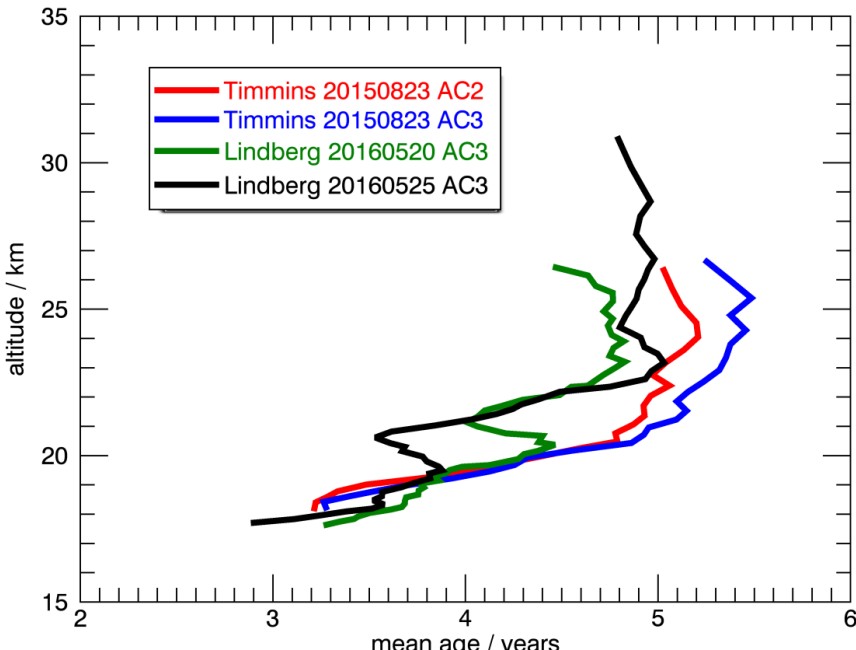

2 Figure 11: Vertical profiles of $CO_2$ derived mean age for the AirCore observations by Univer-

3 sity Frankfurt in 2015 in Timmins, Canada and 2016 in Lindenberg, Germany.





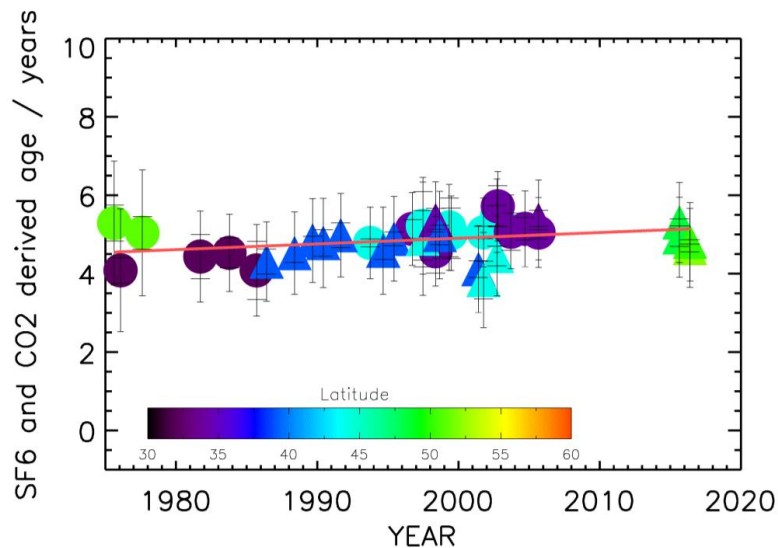

Figure 12: Time series of mean age derived from balloon observations. The data prior to 2010 are those presented in Engel et al. (2009). The data from 2015 and 2016 are derived from the AirCore measurements presented here. Each data point represents the average value of mean age derived above 30 hPa and up to 5 hPa. The inner error bars represent the variability (error of the mean), the larger outer error bars include the uncertainty as discussed in Engel et al. (2009). A non significant trend of 0.15 ($\pm$0.18) years per decade is derived from these observations.





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
