# Peer review of "Mean age of stratospheric air derived from AirCore observations"

_Atmospheric Chemistry and Physics, 2017_

## Referee Comment (RC1) · D. Waugh (Referee) · 3 Mar 2017

This manuscript presents measurements of stratospheric $CO_2$, $CH_4$, and CO together with estimates of the mean age from balloon borne AirCore measurements. This new low cost measurement technique offers the opportunity for more regular measurements of stratospheric $CO_2$ and mean age, which are badly needed to answer questions regarding possible changes in stratospheric age.

It is shown that the mean age can be estimated from AirCore $CO_2$ measurements, although there are some differences between the age profiles from the same flight. These differences (uncertainties in mean age) mean that the measurements presented don't really answer the question of whether the age is increasing or decreasing over long time scales. However, it is very important to show that the age can be estimated

from AirCore measurements, and hopefully many more measurements can be made over coming years that will help to resolve this issue.

The manuscript will be of interest to many ACP readers, is well written, and will be suitable for publication after only a few minor revisions.

SPECIFIC COMMENTS

Pg 2, line 21-23: A strengthening of BDC is expected from model calculations with increasing $CO_2$, and I think you need to include "increasing $CO_2$" in this sentence. Also, I am not sure why you say "should be reflected". The same model calculations with increasing $CO_2$ show a decrease in age (if trend calculated over a long enough time period).

Pg 13, line 6: "Figures"

Pg 19, line 10: There is something missing here "that the For $CH_4$"

Pg 19, line 14: "performed first observation" I think "the" is needed before "first". Also, should it be "observations"?

Figure 12: Why does the y-axis go from 0 to 10, when data is within 2 and 7. I know the authors have published figures with the same scale previously, but I think it would be much better to have a reduced vertical axis, as well as smaller symbols. Then it will be easier to see the values for individual measurements.

---

## Referee Comment (RC2) · E. Ray (Referee) · 14 Mar 2017

This paper describes new measurements of CO2, CH4 and CO taken using the Air-Core technique from two midlatitude sites over several years. These measurements are also used to estimate the mean age of air in the stratosphere in order to extend the mean age time series in the northern midlatitudes. Precise, vertical profile measurements of these trace gases in the stratosphere are rare and valuable for diagnosing the stratospheric circulation. The AirCore technique is a cost-effective means of obtaining these measurements, yet there are many details to consider and this paper is very thorough in describing how the measurements were obtained.

The topic is appropriate for ACP and I would recommend publication with consideration of the minor comments listed below.
Minor comments

Pg. 1, line 20: change "from" to "with"

Pg. 2, line 6: change to "parcel"

Pg. 2, line 8: change "to" to "with"

Pg. 4, line 5: remove "and" before "an"

Pg. 5, lines 14-15: Change sentence to something like "The tubes are joined by solder and light weight adaptors."

Pg. 5, line 17: ". . .tube is open ended."

Pg. 6, line 13: change "stronger" to "more"

Pg. 6, line 14: ". . .is in the wider tube."

Pg. 7, lines 8-9: ". . .for up to 6 hours, which also allowed the instrument to remain heated. . ."

Pg. 7, line 15-16: ". . .allows two lines, which are needed to connect the AirCore for the analysis, to be flushed with a standard. . ."

Pg. 9, line 7: change to "assumption"

Pg. 9, line 10: change to "descend"

Pg. 10, lines 17-18: ". . .will also take into account. . ."

Pg. 11, line 1: add "the" before "starting"

Pg. 11, lines 13-14: ". . .air small, the FG had mixing ratios close to those expected. . ."

Pg. 11, line 15: add comma after "PG"

Pg. 11, line 20: change "where" to "were"

Figures 5-7 and 8-10: would be nice to have these figures side by side to easily compare the features.

Pg. 14, line 11: "In particular, rather small. . ."

Pg. 14, line 29: replace "has been" with "was"

Pg. 14, line 32: ". . .on May 25, which reached a higher. . ."

Pg. 15, line 10: remove "measurement"

Pg. 15, line 11: add "in" after "Timmins"

Pg. 15, line 16: You mention the thermal tropopause here, it might be nice to show this on either a separate plot or combined with one of Figure 8-10. Same with Figure 5-7. And what temperature data did you use?

Pg. 16, line 5: should be consistent with the units here, either pressure altitude or pressure

Figures 5-11: what about error bars on the profiles? It might make it too hard to see features on plots with many profiles but it would be nice to see how the uncertainty in the measurement varies with altitude and from flight to flight. Uncertainty on the mean age profiles would be nice to see as well.

Pg. 17, line 16: change "decrease" to "increase"

Pg. 18, line 32: the Ray et al. (2014) paper seems more relevant to cite here

Pg. 19, line 6: change "using" to "use"

Pg. 19, line 10: ". . .that for CH4. . ."

Pg. 19, line 14: ". . .further performed the first observations from the. . ."

Pg. 20, line 18: change "content" to "context"

Figure 3 caption: change to "flown". Also, may want to put some indication in this

figure of which part of the time series is the lowest vs. highest altitude sample since it's not totally obvious from a quick glance. Maybe also label the white spaces as PG to indicate the push gas measurement and if there is a region where some of the FG is measured that would be interesting as well.

---

## Author Comment (AC1) · 28 Apr 2017

First of all we would like to thank both reviewers for their positive reviews of our manuscript. Both reviewers have raised mainly minor questions which we will answer point by point below. Our answers are shown in italic and changes to manuscript are shown in red.

**Reviewer #1: Darryn Waugh**

This manuscript presents measurements of stratospheric CO2, CH4, and CO together with estimates of the mean age from balloon borne AirCore measurements. This new low cost measurement technique offers the opportunity for more regular measurements of stratospheric CO2 and mean age, which are badly needed to answer questions regarding possible changes in stratospheric age.

It is shown that the mean age can be estimated from AirCore CO2 measurements, although there are some differences between the age profiles from the same flight. These differences (uncertainties in mean age) mean that the measurements presented don't really answer the question of whether the age is increasing or decreasing over long time scales. However, it is very important to show that the age can be estimated from AirCore measurements, and hopefully many more measurements can be made over coming years that will help to resolve this issue. The manuscript will be of interest to many ACP readers, is well written, and will be suitable for publication after only a few minor revisions.

We thank the reviewer for this general positive assessment. There are always uncertainties associated with observations and this is of course true for AirCore observations. We also agree that the new observations do not answer the question of a long-term increase. However, we believe that the uncertainties are sufficiently small that, using a larger number of observations than possible so far, they will in the long term be able to provide a realistic picture of changes in mean age of air.

**SPECIFIC COMMENTS**

Pg 2, line 21-23: A strengthening of BDC is expected from model calculations with increasing CO2, and I think you need to include "increasing CO2" in this sentence.

We added "with increasing greenhouse gas concentrations" to the text, as it is not only CO2.

Also, I am not sure why you say "should be reflected". The same model calculations with increasing CO2 show a decrease in age (if trend calculated over a long enough time period).

This sentences actually mixed models and what would be expected from observations. We have rephrased this to make it clearer that this is the model expectation. We have also added some references to model studies which were missing at this point.

An increase in the strength of the BDC is expected from model calculation with increasing greenhouse gas concentrations (Austin and Li, 2006;Butchart et al., 2006;Butchart, 2014). This is reflected in overall shorter transit times, thus also lower mean age values in the models.

Pg 13, line 6: "Figures"

Changed

Pg 19, line 10: There is something missing here "that the For CH4"

"This indicates that the" has been deleted.

Pg 19, line 14: "performed first observation" I think "the" is needed before "first". Also, should it be "observations"?

Changed to "the first observations"

Figure 12: Why does the y-axis go from 0 to 10, when data is within 2 and 7. I know the authors have published figures with the same scale previously, but I think it would be much better to have a reduced vertical axis, as well as smaller symbols. Then it will be easier to see the values for individual measurements.

The Figure was of course meant to be an update, therefore we kept the same format in the submitted manuscript. However, we have replotted the data with a more narrow range following the reviewer suggestions.

**Reviewer #2: Eric Ray**

This paper describes new measurements of CO2, CH4 and CO taken using the Air-Core technique from two midlatitude sites over several years. These measurements are also used to estimate the mean age of air in the stratosphere in order to extend the mean age time series in the northern midlatitudes. Precise, vertical profile measurements of these trace gases in the stratosphere are rare and valuable for diagnosing the stratospheric circulation. The AirCore technique is a cost-effective means of obtaining these measurements, yet there are many details to consider and this paper is very thorough in describing how the measurements were obtained. The topic is appropriate for ACP and I would recommend publication with consideration of the minor comments listed below.

Thank you for this positive feedback.

**Minor comments**

We have followed all the minor suggestions concerning wording mentioned by the reviewer, unless indicated in the following. .

Pg. 1, line 20: change "from" to "with"

Pg. 2, line 6: change to "parcel"

Pg. 2, line 8: change "to" to "with"

Pg. 4, line 5: remove "and" before "an"

Pg. 5, lines 14-15: Change sentence to something like "The tubes are joined by solder and light weight adaptors."

Pg. 5, line 17: "tube is open ended."

Pg. 6, line 13: change "stronger" to "more"

**Changed to "faster"**

Pg. 6, line 14: " is in the wider tube."

Pg. 7, lines 8-9: "for up to 6 hours, which also allowed the instrument to remain heated

•••

Pg. 7, line 15-16: "allows two lines, which are needed to connect the AirCore for the analysis, to be flushed with a standard

Pg. 9, line 7: change to "assumption" Pg. 9, line 10: change to "descend" Pg. 10, lines 17-18: "will also take into account" Pg. 11, line 1: add "the" before "starting" Pg. 11, lines 13-14: " air small, the FG had mixing ratios close to those expected

Pg. 11, line 15: add comma after "PG" Pg. 11, line 20: change "where" to "were"

Figures 5-7 and 8-10: would be nice to have these figures side by side to easily compare the features.

Thank you for this suggestion. However, we think that the Figures might either become too small or, when including everything in one plot too busy. We believe that including the vertical line to indicate the altitude of the tropopause adds some guidance to the plots, which will allow for better comparision.

Pg. 14, line 11: "In particular, rather small
Pg. 14, line 29: replace "has been" with "was"
Pg. 14, line 32: on May 25, which reached a higher
Pg. 15, line 10: remove "measurement"
Pg. 15, line 11: add "in" after "Timmins"

Pg. 15, line 16: You mention the thermal tropopause here, it might be nice to show this on either a separate plot or combined with one of Figure 8-10. Same with Figure 5-7. And what temperature data did you use?

The thermal tropopause has been calculated using the WMO definition. The data used where from radiosondes flown on the same balloons. We have no included the tropopause altitude as a dashed line in all plots mentioned. Especially in the case of CO there is an excellent agreement between the tropopause altitude and the sharp decrease of CO mixing ratios. This is also mentioned in the text and the explanation to the tropopause altitude is included in all Figure captions.

**Added to Figure captions:**

The dashed lines represents the thermal tropopause according to the WMO definition.

**Changed in text:**

For all species there is a distinct change at the tropopause, which was observed around 10.4 km altitude on May 20 and 10.9 km on May 25. The decrease in tracer mixing ratios, especially for CO, is observed at the same altitude as the thermal tropopause, showing that the altitude attribution as explained in section 2 yields realistic results.

Pg. 16, line 5: should be consistent with the units here, either pressure altitude or pressure

**Changed to pressures below 30 hPa (approximately altitudes above 24 km)**

Figures 5-11: what about error bars on the profiles? It might make it too hard to see features on plots with many profiles but it would be nice to see how the uncertainty in the measurement varies with altitude and from flight to flight. Uncertainty on the mean age profiles would be nice to see as well. The uncertainty of the measurements does not change from campaign to campaign, as long as the performance of the analyzer remains the same. We found standard deviations when measuring our standard of 0.025 ppm for CO2, 5 ppb for CO and below 0.2 ppb for CH4. As these pure analytical precisions are smaller than the thickness of the line (with the exception of CO) it would not make sense to include them in the plots. Instead we have chosen to extend the short description of the analytical precision in the text.

Added in section on overall concept on p.4

Typical reproducibilities observed during field operations showed in precisions of 0.025 ppm of  $CO_2$  and 0.2 ppb of  $CH_4$ . For CO, which was mainly used to distinguish between ambient air and PG, typical precision was 5 ppb.

The same argument applies to the mean age values. These are not determined by analytical precision. We have also added a paragraph on this in the chapter on mean age, at the end of section 4.1 on vertical profile observations.

Note that the accuracy of the mean age values determined here is not limited by the analytical precision of the Picarro analyser, which is typically 0.025 ppm, which is less than a week when translated into mean age.

Pg. 17, line 16: change "decrease" to "increase"

Thank you, of course this was a mistake.

Pg. 18, line 32: the Ray et al. (2014) paper seems more relevant to cite here

This was actually a mistake and has been changed. I wanted to refer to the 2014 paper.

Pg. 19, line 6: change "using" to "use"

Pg. 19, line 10: "that for CH4

- Pg. 19, line 14: "further performed the first observations from the
- Pg. 20, line 18: change "content" to "context"

Figure 3 caption: change to "flown". Also, may want to put some indication in this figure of which part of the time series is the lowest vs. highest altitude sample since

it's not totally obvious from a quick glance. Maybe also label the white spaces as PG to indicate the push gas measurement and if there is a region where some of the FG is measured that would be interesting as well.

Instead of indicating this in the Figure, we would prefer to explain this in the Figure Caption. There is no region where the standard is measured in this plot. We prefer not to add an additional plot for standard measurements, in order not to increase the number of plots further.

We have added the following to the Figure caption:

The grey shaded areas denote the measurements of air from the AirCore. The stratospheric part of the profiles is always measured first. Before the measurements of AC2, between the measurements of the AirCores and after the measurement of AC3, PG is measured by the Picarro analyzer.